# A joint modeling approach for longitudinal microbiome data improves ability to detect microbiome associations with disease

**Pamela N. Luna**[1,2], **Jonathan M. Mansbach**[3], **Chad A. Shaw**[1,2] *

**1** Department of Molecular and Human Genetics, Baylor College of Medicine, Houston, Texas, United States of America, **2** Department of Statistics, Rice University, Houston, Texas, United States of America, **3** Department of Pediatrics, Boston Children's Hospital, Harvard Medical School, Boston, Massachusetts, United States of America

* cashaw@bcm.edu

**Data Availability Statement:** The pregnancy microbiome dataset published by Zhang, et al. (https://doi.org/10.3389/fmicb.2018.01683) is directly available at https://github.com/abbyyan3/

## Abstract

Changes in the composition of the microbiome over time are associated with myriad human illnesses. Unfortunately, the lack of analytic techniques has hindered researchers' ability to quantify the association between longitudinal microbial composition and time-to-event outcomes. Prior methodological work developed the joint model for longitudinal and time-to-event data to incorporate time-dependent biomarker covariates into the hazard regression approach to disease outcomes. The original implementation of this joint modeling approach employed a linear mixed effects model to represent the time-dependent covariates. However, when the distribution of the time-dependent covariate is non-Gaussian, as is the case with microbial abundances, researchers require different statistical methodology. We present a joint modeling framework that uses a negative binomial mixed effects model to determine longitudinal taxon abundances. We incorporate these modeled microbial abundances into a hazard function with a parameterization that not only accounts for the proportional nature of microbiome data, but also generates biologically interpretable results. Herein we demonstrate the performance improvements of our approach over existing alternatives via simulation as well as a previously published longitudinal dataset studying the microbiome during pregnancy. The results demonstrate that our joint modeling framework for longitudinal microbiome count data provides a powerful methodology to uncover associations between changes in microbial abundances over time and the onset of disease. This method offers the potential to equip researchers with a deeper understanding of the associations between longitudinal microbial composition changes and disease outcomes. This new approach could potentially lead to new diagnostic biomarkers or inform clinical interventions to help prevent or treat disease.

## Author summary

Evaluating how changes in the human microbiome influence the onset of disease could lead to the development of novel approaches for diagnosis and treatment. Although

NBZIMM-tutorial/tree/master/NBMM-longitudinal-temporal-data/. This methodology can be implemented using the development version of rstanarm on GitHub at https://github.com/stan-dev/rstanarm. A tutorial for using this approach is available online at https://pamelanluna.github.io/mbjm-tutorial/.

**Funding:** This work was funded by grant R01 AI108588 (JMM) from the National Institute of Allergy and Infectious Diseases (https://www.niaid.nih.gov/). The funders had no role in study design, data collection and analysis, decision to publish, or preparation of the manuscript.

**Competing interests:** The authors have declared that no competing interests exist.

various methods exist to determine significant differences in the microbial compositions between disease outcomes, no methods exist to measure how much changes in the microbiome affect disease onset. This deficiency in analytic methods can be attributed to the difficulty of determining associations between time-dependent covariates and time-to-event outcomes in conjunction with unique challenges of microbiome data analysis. Here we propose a new methodology capable of quantifying the effects of longitudinal microbiome data on time-to-event outcomes that overcomes these obstacles, demonstrating its performance and utility via simulation study and application to real data from a case-control study.

This is a *PLOS Computational Biology* Methods paper.

## Introduction

Multiple studies have found differences in microbial compositions among people with various illnesses, including depression, obesity, asthma, and autism spectrum disorder [1–7]. Importantly, the microbiome can fluctuate over time due to diet or other exposures [8–10]. Furthermore, longitudinal studies have shown that changes in the composition of the microbiome over time are associated with disease outcomes [11–14]. Understanding the complex trajectories of different microbes within a community and the relationship of these trajectories to the onset of human disease is important to uncovering the origins of dysbiosis. This enhanced understanding may eventually help researchers develop new methods for diagnosing and treating disease.

Many methods have been developed to find associations between changes in the microbiome and different outcomes [15]. First, cross-sectional analyses compare microbial compositions between phenotypic groups at a single time point and are extended to longitudinal data by contrasting the results across time points [16, 17]. Second, longitudinal regression models determine significant associations between data covariates and taxa abundances over time [18–20]. Third, multiple methods use smoothing splines to determine the time intervals in which microbial compositions significantly differ between phenotypic groups [21–23]. While all of these methods analyze associations between longitudinal microbiome data and an outcome, they do not account for how these changes affect time-to-event disease outcomes. Two methods for determining associations between microbial compositions and event times have been developed [24, 25], but they only examine the microbiome composition at a single time point.

Evaluating associations between time-dependent biomarkers, such as longitudinal microbiome data, and time-to-event outcomes requires a specialized analytic approach. Typically time-to-event models such as the Cox proportional hazards model are used to determine associations between covariates and event times. However, the inclusion of time-dependent biomarkers in a time-to-event model exposes parameter estimates to increased bias due to potential measurement error, imputation of data at event times, or correlation with other covariates and often violates proportional hazards assumptions [26, 27]. A joint modeling approach was developed to address these issues, allowing the incorporation of time-dependent biomarkers as covariates in a time-to-event model [28–31]. This joint modeling method simultaneously estimates a longitudinal submodel for the time-dependent biomarker and an event submodel for the time-to-event outcomes. The event submodel determines associations between the time-dependent biomarker and event times by including their estimated values

from the longitudinal submodel, rather than the observed values, as a covariate [31]. Given that longitudinal microbiome data are time-dependent biomarkers, this joint modeling approach could be used to determine associations between longitudinal microbiome data and time-to-event outcomes. However, microbiome data do not meet the Gaussian assumption of the longitudinal submodel.

Indeed, microbiome data analysis presents unique challenges. Typically, researchers use 16S rRNA gene sequencing or whole-genome shotgun sequencing as the basis for classifying microbes in a sample. This methodology results in a dataset containing counts of each taxon across all samples (microbiome count data). However, the total number of sequence read counts, or library size, varies across samples. This variation is generally recognized as an experimental artifact of the next-generation sequencing procedure and not biologically informative. To address these differences in library sizes, the data are often transformed into relative abundances (microbiome compositional data). This transformation, however, also has limitations. Indeed, microbiome compositional data cannot be analyzed using typical analytic techniques since the data are 1) non-Gaussian and 2) subject to a unit-sum constraint resulting in a simplex sample space [32, 33]. Other data normalization techniques (e.g., edgeR, DESeq2, cumulative sum scaling) have been developed to allow researchers to analyze microbiome data without transforming the data into relative abundances [34–36]. Unfortunately, these normalization methods hinder the interpretability of the resulting statistical models. Another approach to dealing with varying library sizes is to rarefy the data (i.e., subsample sequence read counts so the total number of read counts is consistent across all samples). While rarefying microbiome count data has become a common approach, this methodology reduces statistical power by discarding useful sample data and thus, results in less precise models [37]. More recent analytic approaches have turned away from the Gaussian distribution and instead directly model microbiome count data using discrete probability distributions, such as the Dirichlet multinomial distribution [38] or negative binomial distribution [18].

We hypothesize that a direct methodological extension of the joint model which accounts for the discrete nature of microbial abundances and the variation in library sizes will identify quantitative associations between longitudinal microbiome data and time-to-event outcomes. In turn, these methodological contributions lead to improved sensitivity and specificity in determining time-to-event outcomes influenced by microbial composition changes. To evaluate this hypothesis we develop a joint modeling framework with its longitudinal submodel formulated as a negative binomial mixed effects model that includes an offset term to adjust for library size. We additionally introduce a parameterization that represents the estimated longitudinal submodel values as scaled relative abundances in the event submodel to address the proportional nature of microbiome data [39] and to improve model interpretability. We then outline how to simulate event times associated with longitudinal microbiome data and apply our joint modeling approach to simulated datasets to illustrate its improved performance over existing alternative methods. Finally, we demonstrate the utility of this methodology by quantifying a previously detected association between longitudinal *Prevotella* abundances in the vaginal microbiome during pregnancy and earlier delivery times [40].

## Methods

### The joint model for longitudinal microbiome count data

The joint model for longitudinal and time-to-event data (joint model) determines associations between endogenous time-dependent covariates and event times [27]. The joint model accomplishes this goal by using a longitudinal submodel to model the time-dependent covariate and then incorporating those model values into the time-to-event model. We extended this joint

modeling approach to appropriately model unrarefied microbiome count data in the longitudinal submodel and incorporate the model values into the time-to-event submodel in a way that allows for interpretable results.

**Longitudinal submodel.**   We modified the longitudinal submodel of the joint model to model subject-specific taxon abundances over time. We analyze taxon abundances in the form of unrarefied sequence read counts, which are non-Gaussian and overdispersed. Rarefying sequencing data essentially subsamples the counts so that the total number of sequence reads in each sample is the same. This throws away potentially useful data and decreases the power of analyses. Although it would be possible to use transformed relative abundances in the joint model assuming a Gaussian distribution, relative abundances often do not follow a Gaussian distribution even after performing common transformations.

To appropriately represent this overdispersed count data, we use a negative binomial distribution. For subject $i$ with sample $j$, we assume the abundance of a single taxon in a sample $y_{ij}$ follows a negative binomial distribution with probability mass function given in Eq 1. This parameterization of the negative binomial distribution has expected value $E[y_{ij}] = \mu_{ij}$ and variance $Var(y_{ij}) = \mu_{ij} + (\mu_{ij}^2 / \theta)$. The shape parameter $\theta > 0$ ensures that $Var(y_{ij}) > E[y_{ij}]$ and controls the amount of overdispersion in the distribution.

$$P(Y = y_{ij}) = \frac{\Gamma(y_{ij} + \theta)}{y_{ij}!\Gamma(\theta)} \cdot \left(\frac{\theta}{\mu_{ij} + \theta}\right)^{\theta} \cdot \left(\frac{\mu_{ij}}{\mu_{ij} + \theta}\right)^{y_{ij}} \tag{1}$$

We model the subject-specific taxon abundances over time using a negative binomial linear mixed effects model. The linear predictor with log link function for the $j^{th}$ sample for subject $i$ at time $t$ (Eq 2) has fixed effects $\beta$ for covariates $x_{ij}(t)$ and random effects $b_i \sim \text{MVNormal}(0, D)$ for covariates $z_{ij}(t)$. To account for the varying library sizes across samples, we introduced an offset variable into the linear model representing the log of the total number of sequence reads in a sample $C_{ij}$.

$$\eta_{ij}(t) = \log(\mu_{ij}(t)) = x_{ij}(t)^T \beta + z_{ij}(t)^T b_i + \log(C_{ij}) \tag{2}$$

To represent the subject-specific abundances, we include the subject as the random intercept covariate. We ensure the time variable is included as a fixed or random effect to analyze the abundances over time.

**Event submodel.**   The original implementation of the joint model determines associations between the linear predictor values $\eta_{ij}$ for the time-dependent covariate and the time-to-event. In the case of the negative binomial linear mixed effects model, the linear predictor is the log of the expected sequence read counts for a given sample. However, the event submodel also needs to account for the total number of sequence reads in a sample. Rearranging Eq 2 shows how we can easily determine the predicted relative abundances using the linear predictor.

$$\frac{\mu_{ij}(t)}{C_{ij}} = \exp\left(x_{ij}(t)^T \beta + z_{ij}(t)^T b_i\right) \tag{3}$$

We include these relative abundance values (Eq 3) in the hazard function for the event submodel (Eq 4) to determine the effect size $\alpha$ between the relative abundance of the taxon and the time-to-event. The hazard function has baseline hazard $h_0(t)$ and effect sizes $\gamma$ for covariates $w$. The hazard function uses the entire longitudinal history up to time $t$, $\mathcal{M}_i(t)$.

$$h_i(t|\mathcal{M}_i(t), w_i) = h_0(t) \exp\left(\gamma^T w_i + \alpha \cdot \phi \cdot \exp\left(x_i^T(t)\beta + z_i^T(t)b_i\right)\right) \tag{4}$$

The parameter $\alpha$ represents the increase in the expected log hazard of disease onset for each one unit increase in relative taxon abundance. However, a unit increase in relative abundance indicates going from 0% to 100% abundance of the taxon, which is uncommon. Therefore, the model may not be able to determine the effect size if the relative abundances are very small and do not show a unit increase. To improve the performance and interpretability of the model, we incorporated a scaling factor $\phi$ for these relative abundance terms. The scaling factor allows for flexibility in the model depending on the types of relative abundances and abundance changes in the data. Using $\phi = 10$ will make the unit a 10% change in abundance, and using $\phi = 100$ will make the unit a 1% change in abundance.

The microbiome joint model simultaneously estimates the two submodels with shared fixed effects $\beta$ and random effects $b_i$ parameters.

**Software implementation.**   This methodology can be implemented using the `rstanarm` R package, which provides tools for Bayesian statistical inference of applied regression models, including the joint model for longitudinal and time-to-event data [41, 42]. We have extended the joint modeling `rstanarm` software to provide the functionality necessary to apply this approach. Code for replicating our analyses is included in supplemental file S1 Code. Furthermore, a tutorial for preprocessing and analyzing data using our methodology is available online and is also included here as S1 Appendix.

## Simulation study

**Longitudinal microbiome count data.**   We simulated the taxon abundances for multiple microbiome samples for each subject over time. We modeled the association between the microbial abundances and sample covariates using the model structure given in Eq 5. The fixed effects include a binary time-independent sample covariate $X_1$, the continuous time variable $t_{ij}$, and their interaction. To emulate the taxon abundance trajectories for each subject, we also included a subject-specific random intercept and a random slope based on the time variable.

$$Y \sim X_1 + Time + X_1 : Time + (Time \mid ID) + \text{offset}(\log(Counts)) \tag{5}$$

We assume the taxon abundances $Y$ follow a negative binomial distribution. Using the log link function, Eq 6 gives the linear predictor for the $j^{th}$ sample from subject $i$.

$$\eta_{ij} = \beta_0 + \beta_1 X_{1ij} + \beta_2 t_{ij} + \beta_3 X_{1ij} t_{ij} + b_{0i} + b_{1i} t_{ij} + \log(C_{ij}) \tag{6}$$

We set the parameter values for the fixed effects $\beta_1$, $\beta_2$, and $\beta_3$. We sampled the random effects $b_i \sim \text{Normal}_2(0, D)$, where $D$ is the variance-covariance matrix with $d_{11} = \text{Var}(b_0) = 0.003$, $d_{22} = \text{Var}(b_1) = 0.001$, correlation parameter $\rho$, and covariance term $d_{12} = d_{21} = \rho \cdot d_{11} d_{22}$.

We then determined the model covariates by randomly sampling $N$ values for the time-independent covariate $X_{1i} \sim \text{Bernoulli}(0.5)$ and $N \times K$ values for the time covariate $t_{ij} \sim \text{Uniform}(0, 8)$. We assumed the total number of sequence reads for each sample followed a normal distribution with $C_{ij} \sim \text{Normal}(10000, 1000)$.

Using these simulated covariates and parameter values, we then evaluated the linear predictor $\eta_{ij}$. However, because $\mu_{ij}$ should give values representing relative abundances, we must restrict its range to $\mu_{ij} \in [0, 1]$. Noting that the intercept term $\beta_0$ scales $\mu_{ij}$ multiplicatively (Eq 7), we initially set $\beta_0 = 0$ when calculating $\eta_{ij}$. We then set $\beta_0 = -\max(\eta_{ij} + \epsilon)$ and recalculated

$\eta_{ij}$ and $\mu_{ij}$ using the new value for $\beta_0$.

$$
\begin{aligned}
\mu_{ij} &= \exp(\eta_{ij}) \\
&= \exp(\beta_0 + \beta_{-0}^T X_{ij} + b_i^T Z_{ij} + \log(C_{ij})) \\
&= \exp(\beta_0) \cdot \exp(\beta_{-0}{}^T X_{ij} + b_i^T Z_{ij} + \log(C_{ij}))
\end{aligned}
\tag{7}
$$

Finally, the longitudinal abundances $Y_{ij}$ were determined by taking $N * K$ samples from NegativeBinomial($\mu_{ij}, \theta$), where $\theta$ is the dispersion parameter.

**Event times.** Generally, event times can be simulated using the event function $S(t) = \exp(-H(t))$, where $H(t) = \int_0^t h(u)du$ is the cumulative hazard function for $h(t)$, by applying the probability inverse transform. The cumulative hazard function is determined by evaluating the integral of the hazard function from 0 to $t$. However, the integral over the hazard function for the joint model for microbiome count data (Eq 8) is intractable.

$$
h(t|\mathcal{M}(t), w) = h_0(t) \exp\left(\gamma^T w + \alpha \cdot \phi \cdot \exp\left[x^T(t)\beta + z^T(t)b\right]\right).
\tag{8}
$$

Crowther and Lambert present a solution for generating event times in instances where the hazard function cannot be integrated analytically to determine a cumulative hazard function [43]. Briefly, the method derives an approximation for the cumulative hazard integral by using Gaussian quadratures. Once the cumulative hazard is calculated, a root finding procedure is then applied in order to solve for the event time $t$. To simulate microbiome joint model event times we apply this methodology, which is implemented in the `simsurv` R package [44].

For the event submodel, we extended the hazard function for a Cox proportional hazards model with covariates $W_1$ and $W_2$. The hazard function for this joint model (Eq 9) incorporates the model values $\mu_{ij}$ from the longitudinal submodel scaled by $\phi = 10$ with effect size $\alpha$.

$$
h_i(t) = \lambda \exp\left(\gamma_1 W_{1i} + \gamma_2 W_{2i} + \alpha \cdot \phi \cdot \exp\left(\beta_0 + \beta_1 X_{1ij} + \beta_2 t_{ij} + \beta_3 X_{1ij} t_{ij} + b_{0i} + b_{1i} t_{ij}\right)\right)
\tag{9}
$$

We assumed an exponential baseline hazard $h_0 = \lambda = 0.1$ for the simulated event times. After setting the parameters $\gamma_1$ and $\gamma_2$, we sampled variables $W_1 \sim$ Bernoulli(0.5) and $W_2 \sim$ Bernoulli(0.3). The parameters and covariates for both the longitudinal and time-to-event submodels were then used by the `simsurv` R package, to simulate event times for the hazard function (Eq 9). Once the event times are simulated, all longitudinal observations after a subject's event time are removed from the dataset for time-to-event analysis. The simulated event times are right censored at $t_{max} = 10$. Example R code for simulating event times associated with microbiome count data is included in supplemental file S1 Code.

## Results

### Model overview

We developed a joint modeling framework to determine associations between longitudinal microbiome count data and time-to-event outcomes that accommodates the distribution of microbiome data while still respecting the inherent proportional characteristic of the microbiome. The model, outlined in Fig 1, consists of a longitudinal and a time-to-event submodel.

The longitudinal component models taxon abundance over time using a negative binomial distribution. The longitudinal model structure addresses the issue of varying library sizes by incorporating an offset variable of the log number of sequence reads for each sample. The model values from the longitudinal submodel are incorporated into the time-to-event submodel in the form of scaled relative abundances. The scaling of the relative abundances improves detection and interpretability of the effect sizes.

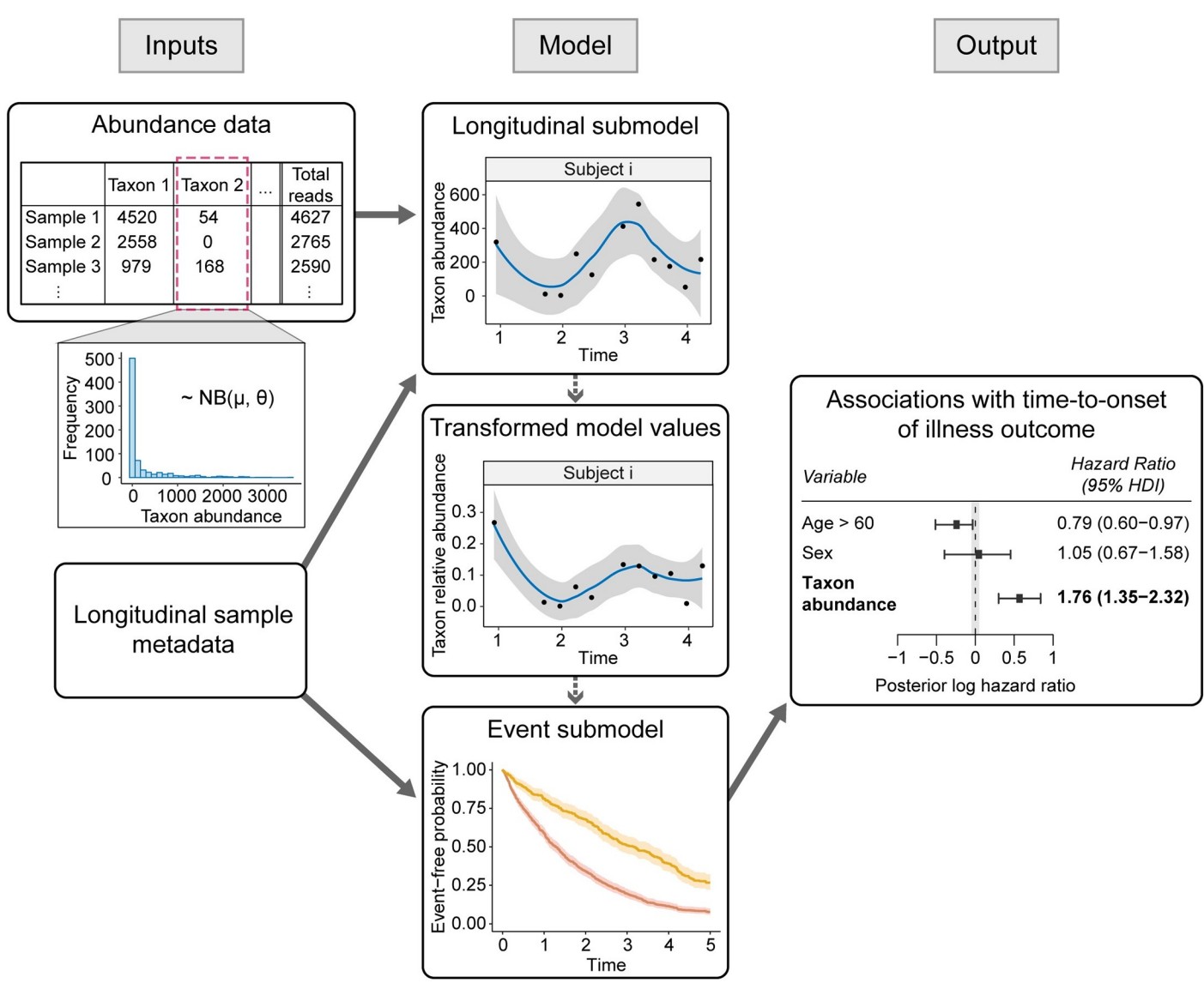

**Fig 1. Overview of joint model for longitudinal microbiome count data.** The inputs, model structure, and output of the joint model for longitudinal microbiome count data. (Inputs) The taxa abundance table contains the sequence read counts for all taxa across samples. The abundances for a single taxon following a negative binomial distribution and the total reads for each sample are passed to the longitudinal submodel. Additionally, metadata for the longitudinal microbiome samples are passed to both the longitudinal and time-to-event submodels. (Model) The longitudinal submodel analyzes subject-specific taxon abundances over time using a negative binomial mixed effects model. The model values for the taxon abundance are transformed to relative abundances before being included in the event submodel. The event submodel determines associations between longitudinal sample data, including the taxon abundances, and the time-to-event for an outcome. (Output) Parameter estimates from the joint model quantify the associations between the time-to-event and model covariates via hazard ratios.

The parameter estimates from the joint model quantify the effect of microbial abundances on the time-to-onset of disease. These parameter estimates can then be used to determine posterior predictions for the longitudinal and time-to-event submodels. Additionally, the joint model's event-free probability predictions can be updated as more longitudinal data is included in the model.

## Simulation study

To assess model performance on data with known parameter values, we analyzed simulated data for $N = 1000$ subjects over $K = 10$ time points. This simulation analysis illustrates that our

model accurately estimates parameter values. While no other methodology exists with the direct aim of quantifying the associations with microbiome data, we show that the joint model performs better than existing alternatives.

**Model performance.**    We applied the joint model for longitudinal microbiome count data to a simulated dataset with taxon abundance effect size of $\alpha = 0.5$. The posterior high density intervals (HDIs) for the longitudinal and time-to-event parameter values are shown in Fig 2A, with true parameter values denoted by the vertical dotted line. For both the longitudinal and time-to-event submodels, the true parameter values fall within their respective 95% HDIs. In particular, the effect size of the taxon abundance parameter $\alpha$ is accurately detected by the joint model, falling in the 50% HDI.

To assess the predictive performance of the microbiome joint model, we predicted the posterior longitudinal trajectories and event-free probabilities using varying amounts of longitudinal data. Fig 2B shows plots of the predicted longitudinal abundances and event-free probabilities using longitudinal data up to $t = 1$ and $t = 4$ split by true event outcome. This figure shows that the model is able to detect the difference in longitudinal trajectories between event outcomes, particularly when predicting the longitudinal trajectory at the later time. Additionally, the model predicts lower event-free probabilities for subjects without the event. This separation becomes more apparent as more longitudinal data is included in the model predictions for event-free probabilities.

**Comparison to alternative methods.**    Although no other methods exist to analyze associations between longitudinal microbiome data and time-to-event outcomes, we were able to compare the performance of our joint model for microbiome count data to existing analytic alternatives. Namely, we compare the performance of our model to the Cox proportional hazards model and the joint model using log-transformed relative abundances. The Cox model does not include the effect of microbial abundances on the time-to-event outcomes. The original formulation of the joint model is the only currently available event model method to include longitudinal microbiome data, but it expects a Gaussian distribution for its longitudinal submodel. To accommodate microbiome data, we normalize the count values to relative abundances and perform a log transformation to shift the data closer to a Gaussian distribution.

Using the same parameter values as above, we simulated event times using various $\alpha$ values and compared the results from these three approaches (Fig 3). The posterior distributions of the model parameters using the moderate $\alpha = 0.5$ taxon abundance effect size are shown in Fig 3A. The parameter estimates for our joint model for microbiome count data always fall in the 95% high density interval (HDI) of the posterior distributions for both the longitudinal and time-to-event submodels. Noting that the Cox model does not have a longitudinal submodel or taxon abundance parameter, we can see that its parameter estimates are close to their true values but that the model compensates for the taxon effect via its baseline hazard. The joint model with transformed relative abundances does have a longitudinal submodel, which has an inaccurately low intercept term. Because the longitudinal model values included in the event submodel are less accurate, the parameter estimates in the event submodel are affected as well. Specifically, the effect of the taxon abundance and the baseline hazard on the time-to-event are both overestimated relative to our method. While we do not expect these models to perform as well as our model since they are not explicitly suited for the simulated data, these results show how analyzing a dataset where this effect is present could skew analytic results.

We also compared the predictive performance of the different models based on the amount of longitudinal data and the true effect size $\alpha$ of taxon abundance. Fig 3B shows the receiver operating characteristic (ROC) curves comparing the ability of the predicted event-free probabilities to differentiate between event outcomes. For $\alpha = 0.1$, the models all have similar

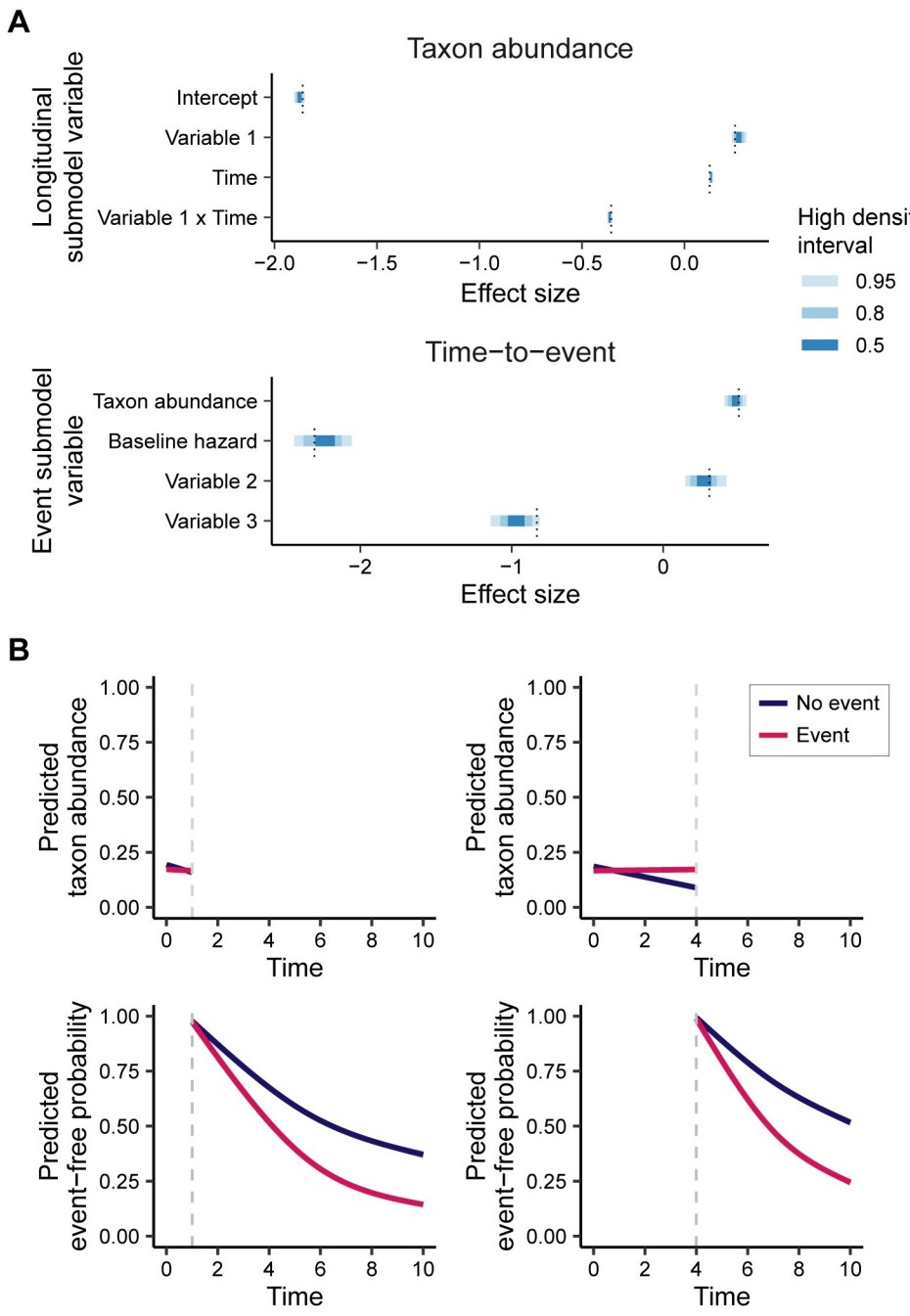

**Fig 2. Model results for simulated data.** Parameter estimates and predictive ability of the joint model for longitudinal microbiome count data on a simulated dataset. (A) Posterior high density intervals (HDIs) for parameters from the longitudinal and time-to-event submodels. Dotted lines show true parameter values. All parameter values fall within the 95% HDIs for the posterior distributions. (B) Marginal predicted longitudinal trajectories and event-free probabilities from the joint model using longitudinal data up to $t = 1$ (left) and $t = 4$ (right) split by true event outcome. As more longitudinal data is provided for the predictions, there is more separation between the marginal event-free probability predictions.

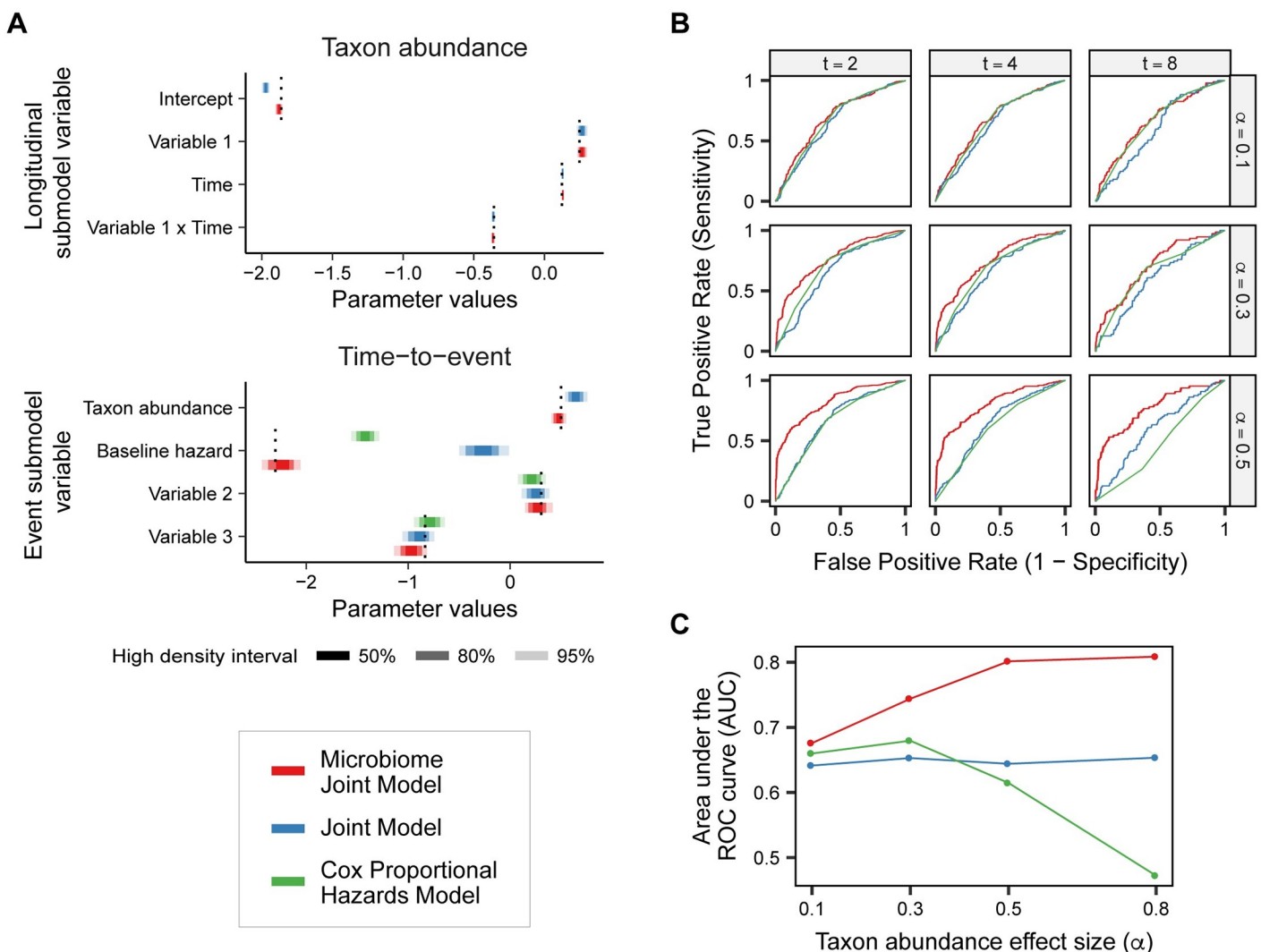

**Fig 3. Model performance compared to alternative methods.** Analysis of the simulated dataset using the joint model for longitudinal microbiome count data, the original joint model with transformed relative abundances, and the Cox proportional hazards model shows that our model best detects relationships within the data. (A) Posterior high density intervals (HDIs) of the parameter estimates for each of the models. The association with taxon abundance is over estimated in the original joint model. Both the Cox model and joint model poorly estimate the baseline hazard, likely accounting for the differences introduced by the effect of the taxon abundance. (B) Receiver operating characteristic (ROC) curves comparing the performance of the event-free probability predictions compared to the true event outcome for the three models. The panels across the x-axis vary the amount of longitudinal data included in the model, and the panels across the y-axis vary the true taxon abundance effect sizes. (C) Comparison of the area under the ROC curves (AUC) for increasing taxon abundance effect size across all three models. As the effect size increases, the performance of the microbiome joint model improves. The microbiome joint model always performs better than the alternative models.

performance since the taxon abundance does not have a large effect on the time-to-event. However, for increasing alpha our model performs successively better than the other models. Additionally, the facets across the x-axis show the model performance using different amounts of longitudinal data to predict the event-free probabilities. As more longitudinal time points are included, the other models perform worse. In fact, when we include longitudinal data up to time $t = 8$ (all longitudinal data), the Cox model actually performs worse than random when the effect size of the taxon abundance is moderate $\alpha = 0.5$. Because the Cox model does not gain any new longitudinal information, its performance only changes with larger values of $t$ since the predicted event-free probabilities are conditioned on not having the event by time $t$.

Looking only at the models with longitudinal data up to time $t = 4$, we compared the area under the ROC curves (AUC) across different taxon abundance effect sizes (Fig 3C). This comparison of the AUCs between the models illustrates that our model always performs better than the alternatives, even for lower values of $\alpha$, and improves for larger values of $\alpha$. The joint model with transformed relative abundances performs about the same regardless of the effect size, while the Cox model predictions deteriorate with larger effect sizes.

**Sample size analysis.** To understand how this methodology performs on datasets of varying size, we examined how the number of subjects $N$ and number of longitudinal samples $K$ affect the model's accuracy in estimating the taxon abundance effect size. For each combination of $N \in \{50, 100, 100\}$ and $K \in \{3, 5, 10\}$, we simulated 100 microbiome joint model data sets using randomly selected parameter values consistently across all combinations. We compared the taxon abundance effect sizes estimated using our methodology to the true parameter values. The results for this performance analysis, shown in S1 Fig, illustrate that the error rates for parameter estimates did not increase dramatically with fewer subjects or longitudinal samples.

## Application to pregnancy dataset

To demonstrate the utility of our methodology, we applied this joint modeling technique to a pregnancy dataset published by Zhang, et al. [19]. The dataset originated from a case-control study on preterm birth outcomes by DiGiulio, et al. [40] which examined the microbiome of various anatomic sites in women throughout pregnancy. For our analysis, we focused on only vaginal swab samples collected from 40 women prior to birth.

DiGiulio, et al. found that women with microbiome profiles with high abundances of *Lactobacillus* were less likely to experience preterm births, defined as delivery before 37 gestational weeks. However, the *Lactobacillus* count abundances did not fit a negative binomial distribution, so *Lactobacillus* was not appropriate for our model. The study also determined a specific microbiome profile containing high amounts of *Prevotella* that had a higher occurrence of preterm births. Looking at the longitudinal *Prevotella* abundances based on preterm outcome (Fig 4A), we found that women who experienced preterm births had higher levels of *Prevotella* throughout pregnancy than those who did not experience preterm births. Therefore, we chose to examine the association between longitudinal *Prevotella* abundances, which follow a negative binomial distribution, and time to delivery outcomes.

Using microbiome samples combined at the genus level, we modeled the longitudinal abundance of *Prevotella* using a generalized linear mixed effects model adjusted for gestational week of collection, history of preterm births, preeclampsia, and race/ethnicity with random slope based on trimester and random intercept by subject. The longitudinal model was offset by the log library size for each sample. The time-to-event component modeled the association between the relative abundance of *Prevotella* and the time to delivery and was adjusted for preeclampsia, race/ethnicity, and income.

The resulting posterior predictions for the parameters of the longitudinal and time-to-event submodels (Fig 4B) show a positive association between longitudinal *Prevotella* abundance and the time to delivery (HR: 1.5; 80% HDI: 1.17-1.97). Because we used a scaling factor of $\phi = 10$, this result indicates that a 10% increase in *Prevotella* abundance is associated with a 1.5-fold increase in hazard of delivery.

## Discussion

We present a discretized extension of the joint model for longitudinal and time-to-event data [30, 42] to evaluate associations between microbial abundances and the onset of disease. As

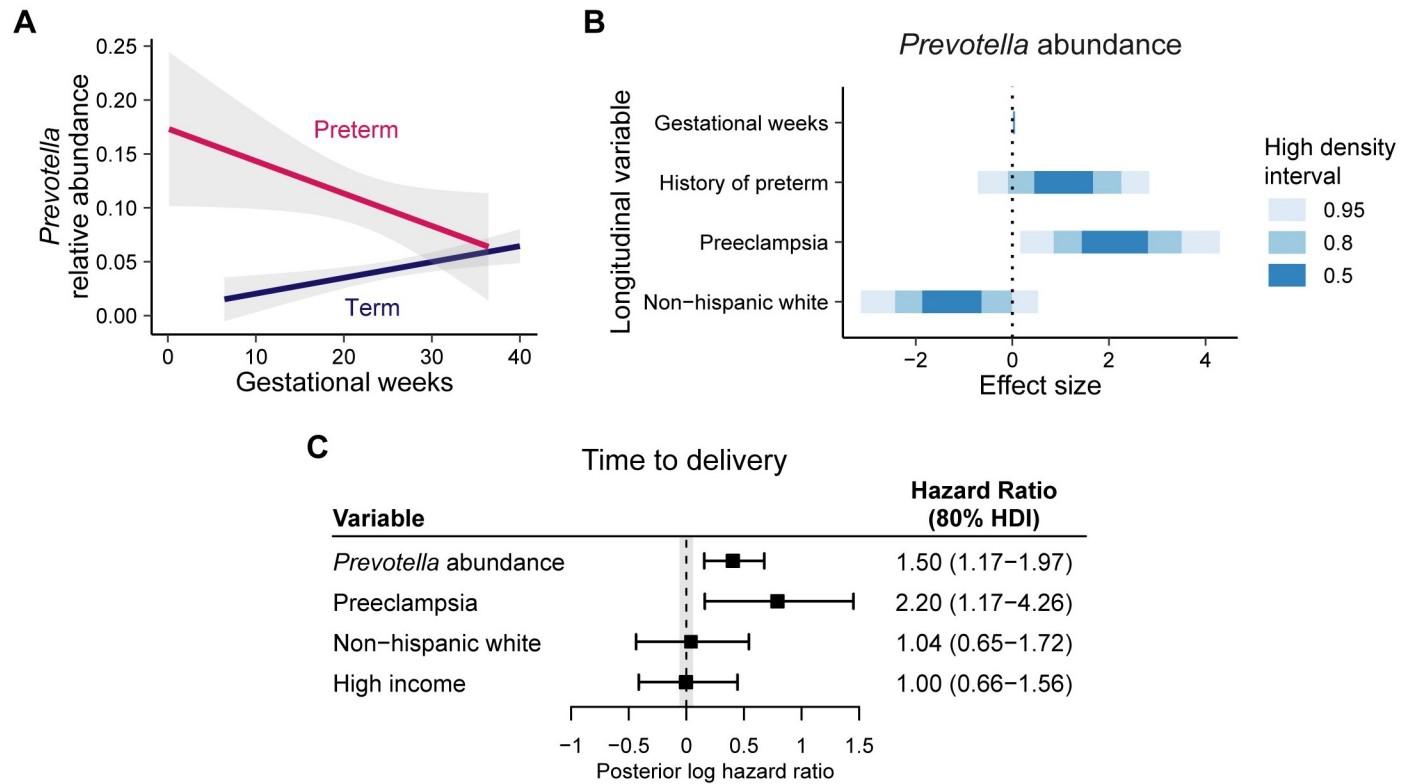

**Fig 4. Analysis of longitudinal pregnancy microbiome dataset.** Joint model for longitudinal microbiome count data analysis of a longitudinal pregnancy microbiome dataset. (A) Observed longitudinal relative abundances of *Prevotella* split by preterm outcome, defined as time to delivery less than 37 gestational weeks. Subjects with a preterm birth outcome initially have higher levels of *Prevotella* that decrease over time. (B) Posterior predictions for the effect sizes of the longitudinal submodel covariates. (C) Posterior predictions of the hazard ratios for the event submodel covariates. The *Prevotella* abundance parameter shows a positive association with the time to delivery outcome.

hypothesized, our approach correctly quantifies associations between longitudinal microbiome data and time-to-event outcomes. Additionally, this joint modelling approach offers improved sensitivity and specificity relative to existing alternative methods in predicting subject-specific event-free probabilities.

In constructing this joint model, we acknowledged the underlying structure of microbiome data by tailoring an existing method to use statistical assumptions appropriate for the data, rather than transforming data to fit the model's assumptions. First, we modified the longitudinal submodel to reflect the characteristics of microbiome count data by using a negative binomial distribution. Second, we included an offset term in the longitudinal submodel that adjusts for the library size of each sample, avoiding the statistically undesirable process of rarefying microbiome data [37]. Third, we parameterized the event submodel to represent microbial abundances estimated by the longitudinal submodel as scaled relative abundances, which addresses the proportional nature of the microbiome [39] and provides interpretable model results.

Using a simulated dataset, we demonstrated that our method accurately models the effects of microbial abundances and other model covariates on time-to-event outcomes. We have also shown the beneficial predictive properties of this model which allow for improved event predictions with additional longitudinal data [31, 42]. Furthermore, we illustrate how this method could be applied in longitudinal microbiome studies via analysis of a pregnancy microbiome dataset [40]. Our results support an association between Prevotella abundance and preterm

birth detected in previous studies [45–49]. However, in addition to reinforcing this finding of Prevotella as a biomarker, we also determined that a 10% increase in the relative abundance of Prevotella indicates a 1.5-fold increase in the hazard of early delivery. This quantification of the relationship between Prevotella abundance during pregnancy and time to delivery is a new result that was unattainable using prior approaches.

Although our novel methodology solves a problem not previously addressed in the field of microbiome research, there remain opportunities for future research in this area. In its current implementation, our model examines the relationship of an individual microbiome taxon and additional covariates with a time-to-event outcome. This approach can be applied in parallel across individual taxa to perform a comprehensive analysis. We recommend using this parallel analysis approach on a methodically selected subset of individual taxa. The Bayesian hierarchical modeling approach utilized in the joint modeling software produces conservative model estimates that obviate the need for multiple testing corrections [50]. An alternative approach is to model the combined longitudinal dynamics and correlations of many taxa at once within the joint modeling framework. Although of interest, the actualization of joint modeling for many taxa is difficult due to the computational complexity of hierarchical Bayesian analyses, where the model complexity grows exponentially in the number of parameters considered [51]. The current implementation of the joint model provides functionality for a multivariate joint model that could model up to three taxa; however, this implementation could violate model assumptions due to the dependency issues intrinsic to microbiome and compositional data [32, 33]. In the future high dimensional Bayesian approaches may enable such model estimation [52].

We argue that the single taxon joint analysis is effective for two reasons. First, we note that clinicians and biologists focus their interest on the largest and most easily interpretable effects —such as the risk impact of individual taxa on outcomes that we demonstrated in the preterm birth application. Higher order effects of many taxa are less interpretable and therefore less actionable. Second, we note that commonly used analytic methods that consider the entire microbial community by clustering data often result in microbiome profiles dominated by a single taxon [40, 48, 53–56]. In these instances, a cluster that is driven by an individual taxon is used in downstream association analyses, which is analogous to our approach.

Another limitation of our model is the restriction of the longitudinal submodel to a negative binomial distribution, which precludes the analysis of taxa with bimodal distributions or excess zeroes. The number of taxa with these distributional complications is often low but can be highly dependent on the microbial diversity within the dataset. Taxa with an overabundance of zero counts might be better modeled using zero-inflated or hurdle models, but these solutions not currently implemented in existing joint model software. In these situations our negative binomial approach could still be applied but with a reduction in statistical power and performance due to lack of model fit. We advise performing preliminary analyses on microbiome data to determine taxa of interest and to ensure model assumptions are satisfied.

Despite its limitations, our methodology could be a powerful tool in understanding the relationship between changes in the human microbiome and disease. While we have discussed this approach in the context of the human microbiome, this joint model is also applicable to general microbiome studies. Furthermore, this analytic method could generally be applied to any dataset with a time-dependent endogenous covariate that follows a negative binomial distribution and that also has time-to-event outcomes.

## Supporting information

**S1 Code. Example R code for analysis and simulation.**
(R)

**S1 Appendix. Tutorial for preprocessing and analyzing data.**
(PDF)

**S1 Fig. Taxon abundance effect size errors using varying sample sizes.** Application of the joint modeling methodology on simulated data sets with varying sizes for the number of subjects (N) and number of longitudinal samples (K) shows that the model retains accuracy with smaller sample sizes. The mean squared errors (MSEs) for the effect size predictions remain low with sample sizes as small as N = 100 with any number of longitudinal samples. The MSEs for effect size predictions are larger with sample size N = 50, but the MSEs are reduced with an increased number of longitudinal samples K.
(TIF)

## Author Contributions

**Conceptualization:** Pamela N. Luna, Jonathan M. Mansbach.

**Formal analysis:** Pamela N. Luna.

**Funding acquisition:** Jonathan M. Mansbach.

**Methodology:** Pamela N. Luna.

**Software:** Pamela N. Luna.

**Supervision:** Chad A. Shaw.

**Validation:** Pamela N. Luna.

**Visualization:** Pamela N. Luna.

**Writing – original draft:** Pamela N. Luna.

**Writing – review & editing:** Jonathan M. Mansbach, Chad A. Shaw.

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
