## [Decision Letter · Decision Letter 0]

10 Mar 2020

Dear Dr. Shaw,

Thank you very much for submitting your manuscript "A joint modeling approach for longitudinal microbiome data improves ability to detect microbiome associations with disease" for consideration at PLOS Computational Biology.

As with all papers reviewed by the journal, your manuscript was reviewed by members of the editorial board and by several independent reviewers. In light of the reviews (below this email), we would like to invite the resubmission of a significantly-revised version that takes into account the reviewers' comments.

We cannot make any decision about publication until we have seen the revised manuscript and your response to the reviewers' comments. Your revised manuscript is also likely to be sent to reviewers for further evaluation.

Sincerely,

Benjamin Althouse

Associate Editor

PLOS Computational Biology

Jason Papin

Editor-in-Chief

PLOS Computational Biology

Reviewer's Responses to Questions

**Comments to the Authors:**

Reviewer #1: The authors developed a joint modeling approach that combines longitudinal microbiome reads data and time-to-event outcomes into a single framework. Results show that the model predictions improve the ones from the limited literature, even thought this comparison might not be completely fair, and not all the models are able to incorporate all data available. The article is very well written with a sound statistical formulation.

Minor concerns:

- In the simulation study, 1000 subjects were used. This would restrict the validity of the conclusions to just the most ambitions projects, since most studies don't have 1000 subjects. It would be interesting to see how the model fairs with fewer data.

- The authors claim that since relative abundances often do not follow a Gaussian distribution, a modelling using a negative binomial distribution would be more appropriate, but as they also noted not all taxa follow this distribution. Some explanation of which assumption is closer to reality would help for clarity.

- I worry about the applicability of the method since it is not well suited for modelling more than a few taxa at a time.

Reviewer #2: This paper presents an approach for identifying disease-associated microbiota members in longitudinal data. My expertise is in software engineering for microbiome research so will provide comments on that topic.

While the modeling approach may be novel (not my expertise) I don't expect to see a lot of use of this approach by the microbiome research community.

First, as the authors note, the model will only be relevant to taxa whose abundance profile follows a negative binomial distribution. The authors advise performing analysis external to their software to determine if the software is applicable to a specific taxon. Most users will either ignore this warning, or choose to use another software package. I recommend that the authors include the test that they feel users should apply as part of their software to determine if assumptions of their approach are violated.

Next, as far as I can tell, there is not a release version of the software, and there is insufficient description of how to apply this approach. I recommend that the authors create an official release of the software, and provide a tutorial illustrating how it can be applied to example data available in a standard file format (e.g., the biom format, which is widely used by microbiome software tools). Without this, microbiome researchers will not know how to apply this method.

If the above comments are addressed, I'll better be able to evaluate this software and provide a prospective as a potential user of the approach.

Finally, it sounds like the user really has to know specifically what microbes they're interested in before this method can be applied (between 1 and 3 community members). It is extremely uncommon for users to have this information in advance in practice.

Line 236: I disagree with this assertion - just because this is what may have been identified in the past, it doesn't mean that interactions between community members are less important.

Reviewer #3: Luna et al. proposed a joint modeling approach for longitudinal microbiome data with survival outcome to detect microbiome associations with disease. The proposed method uses a negative binomial mixed effects model jointing with a Cox hazard model in longitudinal setting with the capability to measure how much changes in the microbiome affect disease onset(survival outcome).

This is very challenge topic in microbiome literature given that the microbiome data are very complicated: structured with tree, compositional, multivariate, sparse and often with many zeros.

Due to the challenge, the development of longitudinal and survival model for analyzing microbiome data are rare. So far only several longitudinal models and two survival models have been proposed. Thus, the proposed method will add a statistical tool to the literature of microbiome methods and provide an alternative to model longitudinal and survival microbiome data. However, the proposed method also has several limitations.

1. The model analyzes one taxon at a time. Actually, the univariate approach cannot account for the compositional nature of microbiome data as claimed by the authors because this approach treats each of taxon as independent and does not consider each taxon in ecosystem.

2. To join the longitudinal negative binomial mixed effects model to survival model, the proposed method introduces “a parameterization that represents the estimated longitudinal submodel values as scaled relative abundances in the event submodel to address the compositional nature of microbiome data and to improve model interpretability.”

Since the proposed method did not use any log-ratio transformation, only scaling the abundant counts into relative abundances could not address the compositional nature of microbiome data. Actually in the simulation study, the relative abundances are assumed to be distributed as normal. Authors need to clarify this confusion.

3. Generally the univariate approach is underpowered compared to multivariate approach and ignores the large p and small n problem. Can the authors comment on how to detect the association between many taxa (e.g., 1000 taxa) and disease onset using the proposed method? Is to test one by one and then adjust for multiple testing?

4. Since the proposed method uses a negative binomial mixed effects model to determine

longitudinal taxon abundances, it is better to provide a procedure to detect the taxa distribution for NB model for guiding the users to appropriately use the method.

5. Since this is methodology paper, the methods are more important, so the Methods Part should be followed by the Introduction part for better presentation.

Minor:

The authors stated that “One method for determining associations between microbial compositions and event times has been developed [24],…”

Actually, we just reviewed, except for reference #24, there has another model for microbiome-based association test for survival outcomes.

Koh, H., A. E. Livanos, M. J. Blaser and H. Li (2018). "A highly adaptive microbiome-based association test for survival traits." BMC genomics 19(1): 210-210.

**Have all data underlying the figures and results presented in the manuscript been provided?**

Reviewer #1: Yes

Reviewer #2: Yes

Reviewer #3: Yes

PLOS authors have the option to publish the peer review history of their article (what does this mean?). If published, this will include your full peer review and any attached files.

Reviewer #1: No

Reviewer #2: No

Reviewer #3: No
---

## [Decision Letter · Decision Letter 1]

14 Jul 2020

Dear Dr. Shaw,

Thank you very much for submitting your manuscript "A joint modeling approach for longitudinal microbiome data improves ability to detect microbiome associations with disease" for consideration at PLOS Computational Biology.

As with all papers reviewed by the journal, your manuscript was reviewed by members of the editorial board and by several independent reviewers. In light of the reviews (below this email), we would like to invite the resubmission of a significantly-revised version that takes into account the reviewers' comments.

Please address Reviewer 2's concerns about software best practices and maintenance.

We cannot make any decision about publication until we have seen the revised manuscript and your response to the reviewers' comments. Your revised manuscript is also likely to be sent to reviewers for further evaluation.

Sincerely,

Benjamin Althouse

Associate Editor

PLOS Computational Biology

Jason Papin

Editor-in-Chief

PLOS Computational Biology

Please address Reviewer 2's concerns about software best practices and maintenance.

Reviewer's Responses to Questions

**Comments to the Authors:**

Reviewer #1: All my concerns have been addressed

Reviewer #2: Again, my assessment is that the new methodology sounds interesting but I have serious concerns about the software.

The authors have partially addressed my concerns about usability of the software through the development of their vignette.

I strongly recommend that the authors make their vignette accessible through their website and setup some kind of continuous integration testing to ensure that the vignette does not become outdated (this would alert them for example if a code change resulted in their vignette not working anymore). As anyone working in bioinformatics knows, it's both common and frustrating to encounter tutorials that are out of date with respect to software. Continuous integration testing of tutorials is really the only effective way to ensure that this doesn't happen.

The repository that this code was forked from now has >60 commits since the authors' initial fork and there is currently a merge conflict that will prevent a merge upstream without first being addressed (not sure how serious the conflict is and whether it impacts the authors' functionality). Will the authors' code be kept up-to-date with the original source? It's possible that one or more of those commits included essential bug fixes which won't be available to your users.

The typical practice for extending a well-maintained software package like rstanarm would be to submit your improvements upstream to the original codebase rather than fork the code and try to maintain a parallel fork. If you're not already in touch with the developers of rstan, just as a word of warning, they might not appreciate your approach. I didn't notice that this was what you were doing on my first review. Forking a project and going your own direction with it is generally considered rude, even though open source licenses (GPL in this case) technically allow it. This is because it can create more work for the original development team. (I apologize if I'm misunderstanding and the authors of this paper are also part of the team developing rstanarm - I may be misunderstanding what is happening here.) It seems that the rstanarm authors have contribution guidelines: http://mc-stan.org/rstanarm/dev-notes/index.html

As a result of the authors' approach of forking a repository but not pushing changes back upstream to the source, the README.md file on their GitHub repository actually doesn't show how to install their software, but rather the upstream software that doesn't contain their changes:

https://github.com/pamelanluna/rstanarm/blob/master/README.md

You're also effectively telling your users to go to the original developers for technical support based on that README file. Again, this is something they might not appreciate (and again, I apologize if I'm misunderstanding something here), especially if they're not supportive of the modifications you made.

The authors mention that their goal is to submit the code back upstream in their response to reviewers ("Because we intend to merge our changes into the existing software..."). In my view this is essential if you want to include anything about this software in the publication.

Reviewer #3: The authors have answered all my questions. The reversion has been significantly improved including presentation, clearity, and discussion of its limitations.

**Have all data underlying the figures and results presented in the manuscript been provided?**

Reviewer #1: Yes

Reviewer #2: Yes

Reviewer #3: Yes

PLOS authors have the option to publish the peer review history of their article (what does this mean?). If published, this will include your full peer review and any attached files.

Reviewer #1: **Yes: **Daniel Ruiz-Perez

Reviewer #2: No

Reviewer #3: No
---

## [Decision Letter · Decision Letter 2]

27 Oct 2020

Dear Dr. Shaw,

We are pleased to inform you that your manuscript 'A joint modeling approach for longitudinal microbiome data improves ability to detect microbiome associations with disease' has been provisionally accepted for publication in PLOS Computational Biology.

Best regards,

Benjamin Muir Althouse

Associate Editor

PLOS Computational Biology

Jason Papin

Editor-in-Chief

PLOS Computational Biology

Reviewer's Responses to Questions

**Comments to the Authors:**

Reviewer #2: The reviewers have addressed my concern. I'm very happy to see their code merged upstream to the parent repository - this represents a significant improvement over the state of the software project during my last review.

**Have all data underlying the figures and results presented in the manuscript been provided?**

Reviewer #2: Yes

PLOS authors have the option to publish the peer review history of their article (what does this mean?). If published, this will include your full peer review and any attached files.

Reviewer #2: No

---

## [Editor Report · Acceptance letter]

4 Dec 2020

PCOMPBIOL-D-19-02214R2 

A joint modeling approach for longitudinal microbiome data improves ability to detect microbiome associations with disease

Dear Dr Shaw,

I am pleased to inform you that your manuscript has been formally accepted for publication in PLOS Computational Biology. Your manuscript is now with our production department and you will be notified of the publication date in due course.

With kind regards,

Livia Horvath
